# Involving a Dedicated Epidural-Caring Nurse in Labor Ward Practice Improves Maternal Satisfaction towards Childbirth: A Retrospective Study

**DOI:** 10.3390/healthcare11152181

**Published:** 2023-08-01

**Authors:** Yun-Han Su, Hsiu-Wei Su, Szu-Ling Chang, Yu-Lian Tsai, Po-Kai Juan, Jen-Fu Tsai, Hui-Chin Lai

**Affiliations:** 1Department of Obstetrics, Gynecology & Women’s Health, Taichung Veterans General Hospital, Taichung 407219, Taiwan; lis0627@vghtc.gov.tw (Y.-H.S.); asus627@vghtc.gov.tw (H.-W.S.); 2Department of Anesthesiology, Taichung Veterans General Hospital, Taichung 407219, Taiwan; youthdear@vghtc.gov.tw (S.-L.C.);; 3Department of Medicine, National Yang-Ming-Chiao-Tung University School of Medicine, Taipei 112304, Taiwan; 4Department of Post-Baccalaureate Medicine, National Chung-Hsin University, Taichung 402227, Taiwan; 5Wuri Lin Shin Hospital, Taichung 414013, Taiwan; 6Show Chwan Memorial Hospital, Changhua 505029, Taiwan

**Keywords:** labor pain, epidural analgesia, dedicated nurse, perinatal care, maternal satisfaction

## Abstract

The quality of healthcare is crucially linked to patient satisfaction, particularly in the provision of neuraxial analgesia for labor pain. Neuraxial analgesia for labor pain control should ideally be readily available when requested. However, in real-world practice, anesthesiologists may not always respond immediately to maternal demands, which can compromise the quality of care. To address this issue, this study aimed to evaluate the effectiveness of involving a dedicated nurse in epidural care to improve maternal satisfaction. This study was conducted in a single tertiary center. Medical records of women with singleton pregnancies above 36 gestational weeks who received neuraxial analgesia for labor pain control were reviewed (N = 354). Among them, 104 women (29%) received care from a dedicated nurse. The results showed that involving a dedicated nurse led to higher maternal satisfaction scores before (4.7 ± 0.5 versus 4.5 ± 0.6, *p* = 0.001), during (4.7 ± 0.6 versus 4.5 ± 0.6, *p* = 0.002), and at 24 h postpartum (4.7 ± 0.5 versus 4.5 ± 0.5, *p* = 0.001), without any adverse impact on maternal, neonatal, or epidural-related complications. These findings suggest that allocating a dedicated nurse to epidural care can effectively enhance maternal satisfaction and potentially improve overall care quality.

## 1. Introduction

Labor pain is significant and probably the most excruciating pain a woman may suffer in her lifetime, which causes immense physiological and psychological stress. When left uncontrolled, labor pain can lead to anxiety, hyperventilation, release of catecholamine, increased peripheral vascular resistance, elevated maternal blood pressure, and decreased placental perfusion [1]. Additionally, excessive pain can make it difficult for women to coordinate uterine contractions and push effectively, ultimately leading to a traumatic childbirth experience that can result in post-traumatic stress disorder, which impairs maternal functioning and mother–infant interactions [2]. Some women may even choose to undergo elective cesarean sections to avoid labor pain [3]. As such, adequate pain management is crucial in obstetric care. The American College of Obstetricians and Gynecologists recommend that labor analgesia should be offered on maternal request, provided no contraindications exist [4]. Ensuring adequate pain relief during labor can lead to a more positive childbirth experience for women.

Over the years, various methods have become available to alleviate labor pain and improve satisfaction for pregnant women. These methods include but are not limited to, systemic opioids [5], inhaled nitrous oxide [6,7], acupuncture [8], and hydrotherapy [9]. Among these options, neuraxial analgesia, which is usually referred to as epidural anesthesia in the delivery room, is considered one of the most effective approaches for reducing pain intensity during labor [10,11]. In most clinical settings in Taiwan, the placement of an epidural catheter is typically performed by a well-trained anesthesiologist. The placement of an epidural catheter is followed by a test dose, which is a small amount of local anesthetic that helps to confirm the catheter’s correct placement, the appropriate level of sensory blockade, and the absence of any immediate adverse effects [12]. Afterward, the anesthesiologist administers analgesics either as a continuous infusion, intermittent bolus upon patient request, or a combination of both [13]. Ensuring the effective and safe administration of neuraxial analgesia requires highly skilled providers and an attentive care team.

While epidural anesthesia remains one of the most-used methods for labor pain control, it is important to acknowledge the potential risks and complications associated with this technique. Maternal hypotension, motor block, and fetal bradycardia are some of the most reported complications during epidural analgesia administration [14]. Additionally, epidural anesthesia has been associated with postpartum complications, such as urinary retention, which can lead to discomfort in women. In rare cases, epidural anesthesia can lead to severe complications such as neuropathy or hematoma [14]. However, it is reassuring to note that despite these potential complications, studies have shown that epidural anesthesia does not increase the risk of cesarean section but may prolong the second stage of labor [15].

In our hospital, a tertiary maternal-fetal center in Taiwan, the administration of labor epidural analgesia is the responsibility of our anesthesia team rather than the labor and delivery staff. However, there have been instances where the anesthesia team was not readily available to respond to maternal requests for epidural insertion or medication adjustments due to other emergency surgeries, particularly during night shifts or holidays. To improve the quality of care and address this issue, a dedicated nurse was recruited to assist with epidural care. The nurse’s duties include counseling patients prior to the procedure, facilitating and assisting with catheter placement, monitoring maternal pain scores, adjusting medication dosage or regimens as needed, administering boluses for breakthrough pain, and managing postpartum complications.

The main objective of our study was to assess whether the involvement of a dedicated nurse in epidural care leads to a significant improvement in maternal satisfaction and physician satisfaction. The second objective of our study was to assess whether the involvement of a dedicated nurse in epidurals leads to any perinatal or epidural impact.

## 2. Materials and Methods

### 2.1. Study Design

This retrospective cohort study was conducted in a single tertiary center. The study population comprised all women older than 20 years old, with singleton pregnancies above 36 gestational weeks who were admitted for vaginal delivery and received neuraxial analgesia between July 2020 and December 2020. A satisfaction survey was given to those women after each delivery. The satisfaction survey was also given to all obstetricians and anesthesiologists in our hospital after the enrollment period. The study was approved by the Institutional Review Board of Taichung Veterans General Hospital with IRB number CE21128A.

### 2.2. Study Materials

The medical records of these women were thoroughly reviewed to obtain information on maternal demographics, obstetric-related outcomes, neonatal outcomes, epidural-related complications, and satisfaction scores related to labor pain management. The satisfaction scores of obstetricians and anesthesiologists were also obtained.

### 2.3. Obstetric Care

In usual obstetric care in our hospital, women who experience the spontaneous onset of labor are admitted if they reach or exceed a 3 cm dilatation of the cervix. Various options for analgesia, including epidural patient-controlled anesthesia (EPCA) and systemic opioids, are offered to them upon admission. In cases where epidural anesthesia is the preferred method, early catheter placement is encouraged before the pain becomes intolerable. However, the timing of catheter placement is ultimately at the discretion of the women themselves. For those who are admitted due to labor induction or premature membrane rupture, we recommend that the epidural catheter is placed before cervical ripening or oxytocin infusion. If these patients decide to receive epidural analgesia, we will administer it regardless of the degree of cervical dilation. We do not discourage women from receiving epidural analgesia during the second stage of labor. Alternative pain management strategies such as water immersion therapy, inhaled nitrous oxide, transcutaneous electric nerve stimulation, acupuncture, or other methods are not available in our hospital.

The epidural anesthesia care team is composed of an anesthesiologist and one or two nurse anesthetists who provide comprehensive care to pregnant women. When women request EPCA placement, the staff of the delivery room notify the anesthesiologist for epidural catheter insertion. In Taiwan, the law stipulates that the technique of epidural anesthesia must be performed by an anesthesiologist. To ensure patient safety and appropriate pain management, a standardized procedure is followed for epidural catheter insertion and medication administration. The test dose is administered immediately after catheter insertion to confirm proper catheter placement. The anesthesia care team stays at the bedside for 30 min to monitor the maternal condition and to check for any adverse reactions. The epidural regimen consists of 0.16% ropivacaine plus 2 μg/mL of fentanyl. The drugs are continuously infused at a basal rate of 10 mL per hour. The patient-controlled bolus setting is 10mL per top-up, with a lockout interval of 15 min and a maximum dose of 120 mL per 4 h. In the event of breakthrough pain, defined as any request for additional epidural medication other than the patient-controlled bolus, 0.5–1% xylocaine 10–12 mL is used as an additional analgesic.

The anesthesiologist and nurse anesthetists are not exclusively dedicated to the EPCA service and may be involved in other routine surgeries or tasks, which could result in delays in responding to women’s needs. In contrast, the dedicated nurse, who is also a trained anesthetist, is exclusively responsible for epidural anesthesia care for pregnant women and does not participate in other surgeries or services during working hours. The dedicated nurse shares responsibilities such as pre-procedure counseling, helping during epidural catheter insertion, ensuring proper catheter function and placement, setting up PCA machines, and educating patients. The dedicated nurse also evaluates maternal pain scores more frequently, usually once every hour (once every 8 h with nurse anesthetists) and is more responsive to demands or complaints. All patients pay the same fee for EPCA service, and they are not charged any additional fee for the dedicated nurse service.

### 2.4. The Studied Intervention Description

The study compared two groups: women who requested labor analgesia, had epidural catheter placement during dedicated nurse working hours, and would receive dedicated nurse service (intervention group) and women who did not have a dedicated nurse involved in these aspects (control group). The dedicated nurse was recruited to work five days a week on the night shift (from 15:00 to 23:00), including one day each weekend. Baseline characteristics such as maternal age, body mass index, ethnicity, parity, gestational age, and comorbidities were compared between the intervention and control groups to identify any significant differences.

### 2.5. Satisfaction Surveys

The satisfaction survey we used in this study is part of a larger questionnaire developed in 2018 at our hospital, assessing patients’ experience while receiving pain management from our physicians and nurses. Five experts reviewed the questionnaire with a Content Validity Index (CVI) of 0.90 [16]. This section of the questionnaire was to evaluate overall satisfaction in three aspects: (1) before epidural catheter insertion, (2) during labor, and (3) after 24 h postpartum. The satisfaction was scored on a five-point Likert scale ranging from 1 (highly dissatisfied) to 5 (highly satisfied). The survey was conducted by an anesthetist who was blinded to the group assignment of the patients at 24 h after each delivery. Additionally, another five-point-scale satisfaction survey was provided to the obstetricians and anesthesiologists who performed the epidural anesthesia after the enrollment period. The surveys for physicians consisted of two questions: (1) what is the level of satisfaction regarding the timeliness of response to patient complaints in both groups? and (2) what is the level of satisfaction regarding the timeliness of response to physician demands in both groups?

### 2.6. Outcome Definition

The main outcome of this study focused on maternal satisfaction scores, as well as the satisfaction scores of anesthesiologists and obstetricians involved in the delivery process. The secondary outcomes of this study included several variables related to labor and delivery. The maternal pain was measured before and after the epidural catheter insertion, specifically 15 min after the epidural analgesia loading dose. A standard visual analogue scale (VAS) was used, with 0 points representing the least pain and 10 points representing the worst pain. The duration of the first and second stage of labor; the incidence of cesarean delivery resulting from prolonged labor or intolerable pain; the time interval between the maternal request for epidural catheter placement; pharmacological analgesia administered before the epidural analgesia, delayed pushing (defined as the interval between full cervical dilation and initiation of maternal pushing of 60 min or more); intrapartum fever (defined as any measurement of the maternal body temperature of 38 degrees Celsius or higher before childbirth); postpartum hemorrhage (defined as blood loss of 500 mL or more for vaginal delivery, and 1000 mL or more for cesarean delivery); neonatal Apgar scores at the first and the fifth minutes; admission to the neonatal intensive care unit (NICU); the requirement of neonatal respiratory support (nasal prong or intubation); and complications associated with neuraxial analgesia, such as intrapartum fever, fetal heartbeat deceleration within 30 min, leg numbness, maternal hypotension, epidural catheter re-insertion, and headache were explored. The attending medical staff recorded these outcomes during the study period.

### 2.7. Sample Selection

The sample size estimation was based on Cheng’s study [17]. We assumed that the satisfaction score would increase by 10% with DN involvement. We conducted a power analysis with a one-sided significant level of α = 0.05 and a power of 1 − β = 0.8. The results indicated that at least 295 people were needed. We enrolled all women who were admitted for vaginal delivery and received neuraxial analgesia via convenience sampling between July 2020 and December 2020. After conducting a thorough examination, we identified a cohort of 354 medical records that pertained to women with singleton pregnancies who were at or beyond 36 gestational weeks and received epidural anesthesia care in our labor wards during the study time. A flowchart depicting the selection process of the study population is presented in Figure 1.

### 2.8. Statistical Methods

The Mann–Whitney U test was used for nonparametric continuous data. Categorical variables were described as a percentage and compared by the chi-square test. Satisfaction scores were tested with Cronbach’s alpha for reliability. We conducted univariate and multivariate linear regression to identify independent factors of the intrapartum satisfaction score as the dependent variate. If a *p*-value in the univariate was <0.2, the variable was included in the multivariate linear regression model. In all analyses, a value of *p* < 0.05 was considered statistically significant. All analyses were performed using SPSS software version 24 (SPSS, Chicago, IL, USA).

## 3. Results

A total of 354 women were enrolled in this study. Among these women, 104 (29%) received epidural care from the dedicated nurse. A summary of the maternal demographics and characteristics is presented in Table 1 to provide a comprehensive overview of the study population. It is noteworthy that no significant differences were observed between the two groups in terms of these demographic and clinical characteristics.

Table 2 summarizes the primary outcomes of the study. The maternal satisfaction scores were found to be significantly higher in the intervention group compared to the control group before epidural injection (4.7 ± 0.5 versus 4.5 ± 0.6, *p* = 0.001), during epidural usage (4.7 ± 0.6 versus 4.5 ± 0.6, *p* = 0.002), and at 24 h postpartum (4.7 ± 0.5 versus 4.5 ± 0.5, *p* = 0.001). The satisfaction scores of the obstetricians were also significantly higher in the intervention group regarding timely response to patients’ complaints (4.7 ± 0.6 versus 2.3 ± 1.0, *p* = 0.004) and timely response to obstetricians’ demands (4.8 ± 0.4 versus 2.7 ± 1.0, *p* = 0.006). The anesthesiologists’ satisfaction scores were significantly higher in the intervention group, regarding timely response to patients’ complaints (4.9 ± 0.3 versus 3.5 ± 1.2, *p* = 0.010) and timely response to the anesthesiologists’ demands (4.8 ± 0.4 versus 3.0 ± 1.4, *p* = 0.006). The intervention group had lower pain scores at the request of EPCA (4.5 ± 2.0 versus 5.1 ± 2.2, *p* = 0.030). Pain scores after epidural injection were similar between the intervention and control groups.

The obstetric and neonatal outcomes of the study are presented in Table 3. There were no significant differences between the intervention and control groups in the duration of labor, the need to delay second-stage maternal push because of EPCA, mode of delivery, indications for cesarean delivery, intrapartum fever, episiotomy, obstetric anal sphincter injury, or postpartum hemorrhage. Furthermore, there were no significant differences observed in neonatal outcomes between the intervention and control groups, including birth weight, Apgar scores at the first and the fifth minutes, admission to the NICU, or requirement for neonatal respiratory support. Among all the enrolled 354 patients, only 2 of them received intravenous opioid analgesia before epidural catheter insertion. Regarding subgroup analysis, we observed that the satisfaction scores revealed no statistically significant differences between nulliparous and multiparous women. Similarly, there were no significant differences in the Apgar scores at the first minute and fifth minute after birth, or for other complications such as respiratory support or NICU admission, between multiparous and primiparous women.

The results regarding epidural anesthesia outcomes and complications are summarized in Table 4. The consumption of epidural analgesics per hour in women who received epidural care from a dedicated nurse versus usual care during labor was similar. However, the intervention group had a significantly longer duration of EPCA usage (19.3 ± 13.8 h versus 14.7 ± 10.7 h, *p* = 0.002), and a higher total consumption of EPCA (131 ± 87 mL versus 162 ± 111 mL, *p* = 0.009). The total care time by the dedicated nurse was significantly longer in the intervention group (6.3 ± 4.1 h versus 3.4 ± 3.7 h, *p* < 0.001). Regarding the safety of epidural analgesia, the study found no significant differences in complications between the two groups, including fetal heart rate deceleration in the first 30 min of EPCA usage, maternal hypotension, epidural catheter re-siting, headache, and leg numbness.

To explore the potential factors affecting intrapartum satisfaction scores (dependent variate), univariate and multivariate linear regression analysis was performed in this study. Univariates included maternal age, maternal BMI, nulliparity or multiparity, induction of labor, cervix dilation at EPCA insertion, pain scores at requesting EPCA, involvement of a DN, EPCA using time, the pain scores before EPCA insertion and during EPCA use, epidural catheter re-insertion, and side effects of PECA. Table 5 summarizes the results of the analyses, demonstrating that pain scores during EPCA use and epidural catheter re-insertion were significantly associated with lower satisfaction scores. The involvement of the dedicated nurse in epidural care was found to be an independent factor contributing to higher intrapartum satisfaction scores.

## 4. Discussion

### 4.1. Results Interpretation

In this retrospective analysis, we demonstrated that involving a dedicated nurse (DN) in epidural care was associated with higher satisfaction scores of the laboring women and the staff, with no difference in obstetric outcomes or analgesia-related complications, and it did not impact the health and well-being of neonates.

Among various options for labor pain control, neuraxial analgesia, specifically epidural analgesia, has been widely recognized as the most effective [13,18]. According to a 2018 Cochrane review, epidural analgesia was found to be superior to non-epidural methods in terms of pain reduction and patient satisfaction [10]. In a randomized clinical trial, only 1% of women allocated to the epidural analgesia group reported negative labor experiences [19]. It is noticeable that all women received epidural analgesia in our study, so it is not surprising that the overall satisfaction scores were remarkably high.

In the present study, it was observed that the intervention group had lower initial pain scores upon requesting epidural analgesia. This observation may be attributed to the involvement of a DN in assisting women with their pain management plans, which may have also provided women with more confidence in their decision-making process, thereby reducing hesitation among women who opted for epidural analgesia during the early stages of labor, when contractions were not yet intolerably painful. Delayed administration of epidural analgesia for more than 15 min was shown to be associated with decreased patient satisfaction in a previous study [20]. Therefore, early catheter placement, which was facilitated by the DN, likely prevented patient dissatisfaction in this study.

Postpartum pain management is a vital part of obstetric care as it can affect a mother’s overall experience of childbirth [5]. After delivery, women typically experience pain during the delivery of the placenta, perineal repair, and uterine massage. The provision of adequate pain relief during this period is critical to ensure patient comfort and satisfaction [5]. In the present study, the DN played a proactive role in encouraging patients to top up an additional bolus of EPCA or continue using EPCA until all these procedures were completed. This approach could have led to a longer duration and higher consumption of EPCA usage in the intervention group. It is noteworthy that despite the higher consumption of EPCA, no significant differences in total labor duration or EPCA-related complications were observed between the intervention and control groups.

The pain score recorded during EPCA use is the subjective measurement for mothers to evaluate the efficacy of labor analgesia. If mothers experience higher pain while using EPCA, they may think the effectiveness of labor analgesia to be insufficient, consequently impacting their overall satisfaction. Catheter re-insertion is a well-known complication of epidural analgesia and can occur for various reasons, including migration of the catheter or displacement of the epidural needle [14]. Our findings suggest that higher pain scores during EPCA use and catheter re-insertion were negatively correlated to maternal satisfaction. The results were consistent with other studies [20,21].

Our study highlights the potential benefits of a DN’s involvement in epidural analgesia care, which not only resulted in increased satisfaction among patients but also improved the work processes of obstetricians and anesthesiologists. Unlike other nurse anesthetists who may be involved in various clinical tasks and surgeries, the DN in our study was able to fully focus on assisting physicians with epidural administration and addressing patient concerns. The DN also shared the workload of anesthesiologists in pre-procedure counseling, ensuring proper catheter function and placement, setting up PCA machines, and educating patients. The anesthesiologists could thus simply perform the epidural catheter placement technique, which significantly reduced their workload. Moreover, the presence of a DN may have helped to mitigate some of the challenges that obstetric care teams commonly face, such as communication breakdowns and task overload. This dedicated support resulted in a smoother workflow for the healthcare team, potentially leading to better patient satisfaction.

Apart from the level of pain reduction, a plethora of factors affect satisfaction with labor pain control [22,23]. Maternal expectation [24] and preference [25] before labor, the level of physical, emotional, and social support [26], the courteous behavior and competency of care providers [27], the availability of human resources and medicines [27], the methods of pain relief [25], a sense of personal control, and maternal and neonatal outcomes [27] have all been proven to be associated with satisfaction. In our labor and delivery ward, the presence of a dedicated nurse who spends more time communicating with laboring women, identifying their needs and problems, and providing support can have a positive impact on maternal satisfaction. Moreover, a dedicated nurse is typically more competent and confident in managing breakthrough pain and complications than general nurses, resulting in a more rapid response to any demand or complaint regarding pain control. Having access to such a service, and knowing that someone can help and will readily help, provides mental support to the often-anxious parturient individuals. In this study, the involvement of a dedicated nurse, who satisfied women’s demands both physically and emotionally, further raised maternal satisfactory levels. To the best of our knowledge, this is the first study to describe the benefit of involving a dedicated nurse in epidural care.

### 4.2. Strengths and Limitations

One of the strengths of this article is that we conducted a satisfaction survey for physicians, either obstetricians or anesthesiologists, conducting a more comprehensive assessment regarding the involvement of a dedicate nurse.

As a retrospective study, it was inherently limited by data availability and the potential for unrecognized confounding factors, and the participants were not randomly assigned to the intervention or control group, which should be acknowledged. Secondly, as the data were collected in a single tertiary medical center in Taiwan, our results might not be reproducible by other obstetric care providers. Thirdly, we only used one section of a validated questionnaire, which may have compromised the validity of this study. Another major limitation of this study is that all babies were delivered by obstetricians in a hospital environment. While the roles of a dedicated nurse and a midwife are not entirely overlapping, the potential benefit of a dedicated nurse may be less significant in maternal care settings where midwives are involved in labor and childbirth. Also, our hospital does not charge patients any additional fees for the dedicated nurse service. Whether to employ a dedicated nurse may vary for each hospital based on factors such as the local average income or hospital surplus. Further studies are needed to explore the generalizability and potential benefits of a dedicated nurse in various maternal care settings.

## 5. Conclusions

Taken together, our results highlight that the presence of a dedicated nurse for epidural care improves the satisfaction of both the women undergoing childbirth and their healthcare providers, without any adverse effects on obstetric or neonatal outcomes. These findings have important implications for the management of labor and delivery and may guide the development of future strategies to improve the quality of care provided to women during childbirth.

## Figures and Tables

**Figure 1 healthcare-11-02181-f001:**
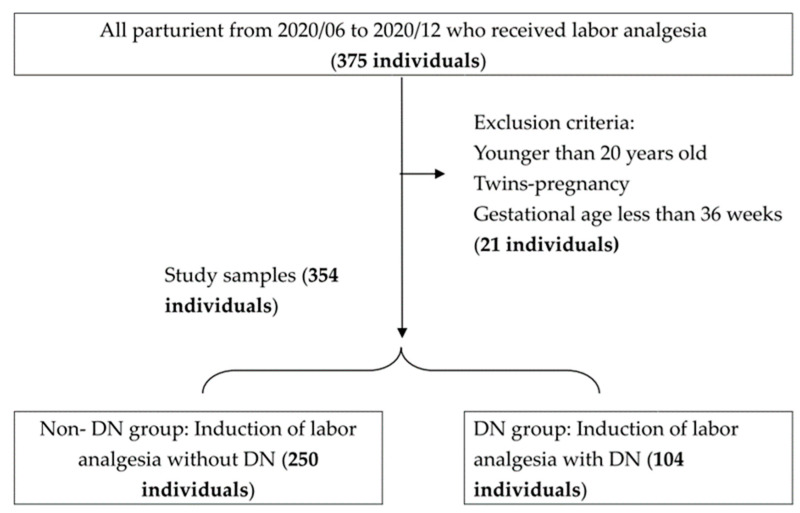
The flowchart of study sample selection.

**Table 1 healthcare-11-02181-t001:** Demographic data of parturients who initiated labor analgesia with or without a dedicated nurse (DN).

Maternal Characteristics	Without DN (N = 250)	With DN Group (N = 104)	*p*-Value
	Mean +/− SD or n (%)	Mean +/− SD or n (%)	
Age (years)	32.6 ± 4.4	32.5 ± 4.7	0.72
BMI (kg/m^2^)	26.5 ± 3.7	27.0 ± 4.8	0.048 *
Gestational age (weeks)	39.0 ± 1.1	39.2 ± 1.1	0.124
Nulliparity †	160 (64%)	68 (65%)	0.804
Gestational Diabetes †	19 (7.6%)	7 (6.7%)	0.775
Pre-eclampsia †	3 (1.2%)	0 (0%)	0.263

Mann–Whitney U; †: Chi-square test; SD: stander deviation; BMI: body mass index; * *p* < 0.05.

**Table 2 healthcare-11-02181-t002:** The satisfaction score of patients, obstetricians, and anesthesiologists between two groups.

	Without DN (N = 250)	With DN Group (N = 104)	*p*-Value
Patients Satisfaction Scores	Mean +/− SD	Mean +/− SD	
Before injection	4.5 ± 0.6	4.7 ± 0.5	<0.001 *
During epidural usage	4.4 ± 0.6	4.7 ± 0.5	<0.001 *
24 h postpartum	4.5 ± 0.6	4.7 ± 0.5	<0.001 *
Satisfaction Score of obstetricians			
Timeliness of response to patient complaints	2.3 ± 1.0	4.7 ± 0.6	0.004 *
Timeliness of response to obstetricians’ demands	2.7 ± 1.0	4.8 ± 0.4	0.006 *
Satisfaction Score of anesthesiologists			
Timeliness of response to patient complaints	3.5 ± 1.2	4.9 ± 0.3	0.010 *
Timeliness of response to anesthesiologists’ demands	3.0 ± 1.4	4.8 ± 0.4	0.006 *

Mann–Whitney U test; * *p* < 0.05.

**Table 3 healthcare-11-02181-t003:** The obstetric and neonatal outcomes between two groups.

	Without DN (N = 250)	With DN Group (N = 104)	*p*-Value
Obstetric Outcomes	Mean +/− SD	Mean +/− SD	
First stage of labor length (min)	422 ± 407	487 ± 471	0.204
Second stage of labor length (min)	101 ± 103	104 ± 95	0.944
Mode of delivery †			
Cesarean section	31 (12%)	14 (14%)	0.948
Forceps or Vacuum	15 (6%)	8 (8%)	0.518
Indication for Cesarean Section †			
Labor dysfunction	24 (10%)	11 (11%)	0.945
Fetal distress	6 (2%)	3 (3%)	0.805
Others	1 (1%)	0 (0%)	
Intrapartum fever > 38.0 °C †	14 (6%)	9 (9%)	0.365
Postpartum hemorrhage †	24 (10%)	15 (15%)	0.244
Neonatal Outcomes			
Apgar score			
First minute	7 ± 1.3	7 ± 1.2	0.397
Fifth minute	9 ± 0.8	9 ± 0.8	0.319
Respiratory support †	12 ( 4.8%)	6 (5.8%)	0.749
NICU admission †	17 (6.8%)	6 (5.8%)	0.670

Mann–Whitney U test; †: Chi-square test; NICU: neonatal intensive care unit.

**Table 4 healthcare-11-02181-t004:** Epidural outcomes between two groups.

	Without DN (N = 250)	With DN Group (N = 104)	*p*-Value
	Mean +/− SD	Mean +/− SD	
Pain score at request	5.1 ± 2.2	4.5 ± 2.0	0.030 *
Pain score after injection	1.8 ± 2.6	1.6 ± 2.5	0.490
Total EPCA consumption (mL)	162 ± 111	131 ± 87	0.009 *
Duration of EPCA (hour)	14.7 ± 10.7	19.3 ± 13.8	0.002 *
EPCA consumption per hour (mL)	9.7 ± 3.5	10.2 ± 6.3	0.360
Total care time of DN (hour)	3.4 ± 3.7	6.3 ± 4.1	<0.001 *
Manual bolus for breakthrough pain			
By DN	0.3 ± 0.6	0.6 ± 1.0	<0.001 *
By ANS	0.3 ± 0.9	0.3 ± 0.7	0.635
FHR deceleration within 30 min †	3 (1.2%)	2 (1.9%)	0.600
Leg numbness †	92 (37%)	47 (45%)	0.120
Maternal hypotension †	43 (18%)	17 (18%)	0.840
Epidural re-insertion †	21 (8.4%)	7 (6.7%)	0.590
Headache †	1 (1.0%)	4 (1.6%)	0.660

Mann–Whitney U test; †: Chi-square test; EPCA: epidural patient-controlled analgesia; FHB: fetal heartbeat; ANS: anesthesiologist; * *p* < 0.05.

**Table 5 healthcare-11-02181-t005:** Univariate and multivariate linear regression analysis for intrapartum satisfactory score.

Variable	Univariate				Multivariate			
	B	B 95%CI (Lower)	B 95%CI (Upper)	*p*-Value	B	B 95%CI (Lower)	B 95%CI (Upper)	*p*-Value
Age	0.005	−0.008	0.017	0.467				
BMI	0.000	−0.017	0.016	0.952				
Nulliparity	−0.76	−0.203	−0.050	0.236				
Multiparity	0.76	0.203	0.050	0.236				
Induction of labor	−0.004	−0.128	0.119	0.943				
Cervix dilation at EPCA injection (cm)	−0.033	−0.069	0.003	0.068 *	−0.030	−0.067	0.007	0.112
Pain score at request EPCA	−0.026	−0.054	0.002	0.065 *	−0.014	−0.043	0.016	0.363
With DN	0.230	0.100	0.360	0.001 *	0.184	0.050	0.319	0.007 **
Total EPCA time (min)	0.000	−0.001	0.001	0.954				
VAS before EPCA	−0.003	−0.019	0.013	0.714				
VAS during EPCA	−0.032	−0.055	−0.009	0.007 *	−0.030	−0.054	−0.006	0.014 **
Epidural re-insertion	−0.409	−0.620	−0.199	<0.001 *	−0.395	−0.613	−0.177	<0.001 **
Headache	−0.519	−1.026	−0.011	0.045 *	−0.360	−0.862	0.141	0.158
Leg numbness	0.041	−0.083	0.165	0.516				
Nausea or vomiting	−0.007	−0.812	0.798	0.986				
Dizziness	−0.517	−1.084	0.050	0.074 *	−0.388	−0.936	0.160	0.165
Pruritis	0.490	−0.643	1.623	0.396				
FHB deceleration within 30 min after injection	0.090	−0.421	0.601	0.730				
First stage of labor (min)	0.000	0.000	0.000	0.732				
Second stage of labor (min)	0.000	−0.001	0.000	0.372				

EPCA: epidural patient-controlled analgesia; VAS: visual analogue scale; * *p* < 0.2; ** *p* < 0.05.

## Data Availability

All data presented in this study are available on request from the corresponding author.

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
