# Peer review of "Involving a Dedicated Epidural-Caring Nurse in Labor Ward Practice Improves Maternal Satisfaction towards Childbirth: A Retrospective Study"

_healthcare, 2023, doi:10.3390/healthcare11152181_

Round 1
Reviewer 1 Report (Previous Reviewer 2)
Congratulations to the authors for this interesting manuscript.
Author Response
Response to Reviewer 1 Comments
Point 1: Congratulations to the authors for this interesting manuscript.
Response 1: Thank you very much the kind words and congratulations on our manuscript. We truly appreciate your encouragement.

Reviewer 2 Report (New Reviewer)
Dear authors,
I hope this message finds you well. I am writing to you regarding your manuscript entitled Involving a Dedicated Epidural-Caring Nurse in Labor Ward 2 Practice Improves Maternal Satisfaction towards Childbirth: a 3 Retrospective Study." which you submitted to HEALTHCARE for possible publication. As a part of the peer review process, I have carefully examined your manuscript in terms of its relevance, scientific rigor, and originality within the context of the research field.
I would like to commend your efforts on addressing a significant issue in this area, and the rigorous research methodology applied in your study. However, I have identified several areas in the manuscript that I believe could benefit from further refinement to strengthen your arguments and enhance the overall clarity and impact of your research.
Please find below my detailed comments and suggested revisions:
Materials and Methods
- As you may know, the indication for epidural analgesia is different in spontaneous labor than in labor induction. Have you documented this variable? I believe it is an important variable when determining a woman's satisfaction with epidural analgesia, as the intensity of pain in induction is usually greater. On the other hand, as you have reflected in the manuscript, the indication for epidural analgesia in active, spontaneous labor has been when women were 3 cm dilated or more; what was the indication for epidural analgesia in those women who underwent labor induction? When did these women receive the epidural?
- You have perfectly defined in the introduction the other analgesia alternatives available to women. My question is whether you have been able to record and document whether both the women in the control group and the experimental group consumed any other type of pharmacological or non-pharmacological analgesia before the epidural analgesia; as in this case, this variable, combined with the previous one, can act as a confounding factor and alter the results and conclusions.
- Did you use normality tests? Do the variables under study have a normal distribution? Why have you used a non-parametric test like the Mann Whitney U in one case, and a parametric test like the Chi-square test in another case?
Results
4. Has a fundamental variable such as parity been taken into account when collecting information and performing statistical analyses? If you have this data, it would be appropriate to include in the statistical analyses the differences in mean satisfaction scores as well as obstetric and neonatal outcomes between primiparous and multiparous women.
Discussion
- There are uncited paragraphs in the discussion. Please insert the corresponding citations, for example in the following paragraph "Postpartum pain management is a vital part of obstetric care as it..."
- In this section, you refer to the importance of the nurse dedicated to the control of epidural analgesia in terms of maternal satisfaction and in terms of obstetric and neonatal outcomes. I would have liked for you to include a variable regarding maternal mobility or leg blockage during epidural analgesia as, as you well know, a complication of the epidural is the blockage of the lower limbs and this can cause a decrease in the woman's mobility as well as therefore hinder position changes and maternal expulsion.
- The following assertion about the study's limitations is completely false: “It is important to acknowledge that retrospective studies are inherently limited by the availability of data and the potential for unrecognized confounding factors. However, we believe that the effect of such factors was minimal in our study as allocation into the intervention or control group was determined by the availability of the dedicated nurse at the time of epidural analgesia request, like randomization.” Claiming this after conducting a retrospective study without any type of sample randomization is totally incorrect. One of the limitations of the study is precisely the sampling of the participants.
These comments are provided in the spirit of constructive criticism with the intention to assist you in enhancing the quality of your manuscript, and ultimately increasing its chances of being accepted for publication in HEALTHCARE; I look forward to reading your revised manuscript.
Kind regards,
Author Response
**Tables are presented in the attachment**
Response to Reviewer 2 Comments
Point 1: As you may know, the indication for epidural analgesia is different in spontaneous labor than in labor induction. Have you documented this variable? I believe it is an important variable when determining a woman's satisfaction with epidural analgesia, as the intensity of pain in induction is usually greater. On the other hand, as you have reflected in the manuscript, the indication for epidural analgesia in active, spontaneous labor has been when women were 3 cm dilated or more; what was the indication for epidural analgesia in those women who underwent labor induction? When did these women receive the epidural?
Response 1: For those who were admitted due to labor induction or premature membrane rupture, we recommended that the epidural catheter be placed before cervical ripening or oxytocin infusion (line 103-104). After explanation, if these patients agree to receive epidural analgesia, we will administer it regardless of the degree of cervical dilation (We added this sentence in line 105-106). The average cervical dilation was 0.8 cm for those patients. We also performed univariate and multivariate linear regression analysis to see if induction of labor be a potential factors affecting intrapartum satisfaction scores, and the result revealed not an independent factor (we added this result in table 5).
Point 2: You have perfectly defined in the introduction the other analgesia alternatives available to women. My question is whether you have been able to record and document whether both the women in the control group and the experimental group consumed any other type of pharmacological or non-pharmacological analgesia before the epidural analgesia; as in this case, this variable, combined with the previous one, can act as a confounding factor and alter the results and conclusions.
Response 2: We recorded all pharmacological analgesia administered before the epidural analgesia in both the computer system and the paper nursing records. However, non-pharmacological analgesia, such as shower, birth ball, acupressure, or relaxing breathing techniques, was not documented. Among all the enrolled 354 patients, only 2 of them received intravenous opioid analgesia before epidural catheter insertion. We clarified this in line 171-172, and line 237-239.
Point 3: Did you use normality tests? Do the variables under study have a normal distribution? Why have you used a non-parametric test like the Mann Whitney U in one case, and a parametric test like the Chi-square test in another case?
Response 3: We performed normality test in all continuous data. We found age, BMI, gestational age, satisfactory scores, 1st stage labor time, 2nd stage labor time were not normal distributed. We performed Mann Whitney U for those variables. The Chi-square test was used for categorical variables. To differentiate these variables, we added a cross mark to them in table 1, table 3, and table 4 to indicate the usage of the Chi-square test.
Point 4: Has a fundamental variable such as parity been taken into account when collecting information and performing statistical analyses? If you have this data, it would be appropriate to include in the statistical analyses the differences in mean satisfaction scores as well as obstetric and neonatal outcomes between primiparous and multiparous women.
Response 4: There were 228 primiparous women, and 126 multiparous women in this study. The primiparous women accounted for 64% in control group (without DN) and 65% in intervention group (with DN), which was in table 1. We performed univariate and multivariate linear regression analysis to see if nulliparity or multiparity be a potential factor affecting intrapartum satisfaction scores, but both of them were not an independent factor (we added this result in table 5). The relationship between parity and satisfaction score; the relationship between parity and maternal-infant outcomes were as following tables. The satisfaction score showed no statistically significant difference in the incidence between the two groups. Primiparous women had significantly longer durations for both the first and second stages of labor compared to multiparous women. Additionally, the rate of cesarean section was significantly higher among primiparous women, which was compatible with our clinical experience. There were no statistically significant differences in the 1st minute and 5th minute Apgar scores, as well as other complications such as respiratory support or NICU admission between multiparous and primiparous women. We added these findings in the “Results” section (line 239-243).
Point 5: There are uncited paragraphs in the discussion. Please insert the corresponding citations, for example in the following paragraph "Postpartum pain management is a vital part of obstetric care as it..."
Response 5: The proper citations were inserted into discussion section (line 303, line 305).
Point 6: In this section, you refer to the importance of the nurse dedicated to the control of epidural analgesia in terms of maternal satisfaction and in terms of obstetric and neonatal outcomes. I would have liked for you to include a variable regarding maternal mobility or leg blockage during epidural analgesia as, as you well know, a complication of the epidural is the blockage of the lower limbs, and this can cause a decrease in the woman's mobility as well as therefore hinder position changes and maternal expulsion.
Response 6: We did record leg numbness as a complication in the medical record. We thought leg numbness might share a degree of similarity with leg blockage. The leg numbness rate was 37% in control group (without DN), and 45% in intervention group (with DN). There is no statistically significant difference in the incidence between the two groups (Table 4). We performed univariate and multivariate linear regression analysis to see if leg numbness be a potential factors affecting intrapartum satisfaction scores, but leg numbness was not an independent factor (Table 5). It is true that maternal mobility may affect labor progreesion, however, we did not record maternal mobility in the initial medical documents. The relationship between leg nubness and C/S rate was as following tables. Whether with leg numbness or not revealed no statistically significant difference in C/S rate.
Point 7: The following assertion about the study's limitations is false: “It is important to acknowledge that retrospective studies are inherently limited by the availability of data and the potential for unrecognized confounding factors. However, we believe that the effect of such factors was minimal in our study as allocation into the intervention or control group was determined by the availability of the dedicated nurse at the time of epidural analgesia request, like randomization.” Claiming this after conducting a retrospective study without any type of sample randomization is totally incorrect. One of the limitations of the study is precisely the sampling of the participants.
Response 7: We revised the paragraph as “As a retrospective study, it was inherently limited by data availability and the potential for unrecognized confounding factors, and the participants were not randomly assigned to the intervention or control group, which should be acknowledged. “(Line 353-355)

Reviewer 3 Report (New Reviewer)
Review for Healthcare, version for Perinatal and Neonatal Medicine
It looks as though I have reviewed a paper which has already undergone review and rewrite.
This study was clearly outlined and organized with very well-defined endpoints. Methodology, including statistical analysis, was well described. The paper itself is well-written with only a few instances of minor grammatical oversights as listed below. Overall, I think the paper makes a very positive contribution to the care of laboring women, as it demonstrates that a nurse dedicated to caring for the patient’s epidural benefits not only the patient in terms of their satisfaction, but also the satisfaction of obstetricians and anesthesiologists. The study describes a model which could seemingly be adopted easily. The only shortcoming is the lack of comments on whether use of a dedicated nurse hospital or patient costs, thus rendering it less likely to be employed widely. Regardless, the paper merits publication.
Line 62: “urinary retention, which can lead to discomfort and discomfort in women.” “Discomfort” is mentioned twice unnecessarily.
Line 176: “neonatal respiratory support (nasal prone or intubation),” Do the authors mean “nasal prongs”? If not, what is “nasal prone?”
Line 243: “The total care time by the dedicated nurse and found that it was significantly longer in the intervention group…” Incomplete sentence. Recommend “The total care time by the dedicated nurse was significantly longer…”
Author Response
Response to Reviewer 3 Comments
Point 1: Line 62: “urinary retention, which can lead to discomfort and discomfort in women.” “Discomfort” is mentioned twice unnecessarily.
Response 1: We deleted the duplicated words, and now the sentence reads “which can lead to discomfort in women.” (Line 62)
Point 2: Line 176: “neonatal respiratory support (nasal prone or intubation),” Do the authors mean “nasal prongs”? If not, what is “nasal prone?”
Response 2: We did mean “nasal prongs”. The mistake was corrected (line 178).
Point 3: Line 243: “The total care time by the dedicated nurse and found that it was significantly longer in the intervention group…” Incomplete sentence. Recommend “The total care time by the dedicated nurse was significantly longer…”
Response 3: We revised the sentence as recommended. Now the sentence reads “The total care time by the dedicated nurse was significantly longer in the intervention group” (line 252-253).
Point 4: The only shortcoming is the lack of comments on whether use of a dedicated nurse hospital or patient costs, thus rendering it less likely to be employed widely.
Response 4: Our hospital allocates approximately 50 thousand New Taiwan Dollars per month (around 1600 US Dollars per month) to employ a dedicated nurse. Patients are not charged any additional fee for the dedicated nurse service. Whether to employ a dedicated nurse may vary for each hospital based on factors such as the local average income, hospital surplus, and other considerations. We clarify this in line 135-137, line 362-365.

This manuscript is a resubmission of an earlier submission. The following is a list of the peer review reports and author responses from that submission.
Round 1
Reviewer 1 Report
Dear authors,
I had the pleasure of reviewing this manuscript which attempts to study the efficacy of having a nurse dedicated to epidural care in the delivery room of a hospital in Taiwan.
I found the topic very interesting. It is necessary to address the pain suffered by the woman during childbirth to improve the experience of labor and delivery and thus avoid other complications caused by the pain and stress suffered by the mother at the time of delivery.
I agree with the authors that having a nurse dedicated to epidural analgesia will surely bring benefits for women's satisfaction, in addition to relieving the workload of midwives, who sometimes have to care for several women at the same time. But the problem, at least in the setting I work in, is that anesthesiologists are always overworked and often don't respond to requests from mothers for epidural analgesia. At this point, we may find ourselves in the position that we have a nurse who will take care of the woman, but we do not have an anesthesiologist willing to perform the analgesic technique to help ease the woman's pain. In the latter case, there is little the epidural nurse can do.
Well, with the sole objective of improving the quality of the manuscript, I will allow myself to make a series of comments:
1. Introduction section.
The introduction seemed adequate to me, but I recommend adding a couple of lines to the last paragraph explaining explicitly what the objective of this investigation was. This point was addressed in the abstract but, however, it was not expressed in the introduction and I consider it important.
2. Material and methods section.
It is necessary to explain in a more developed way which questionnaires were used to measure maternal satisfaction. There are already validated questionnaires in the literature that perfectly measure maternal satisfaction with childbirth. Some of them attend to the pain perceived by the women, to the attention received by the midwife, the obstetrician or the anesthesiologist. It would be necessary to know if some of these validated questionnaires have been used or self-made questionnaires have been used. In this case, it must be explained how the questionnaire was developed and validated to assess its psychometric properties.
3. The choice of participants must appear in the methodology section. The text written between lines 180 and 185 and figure 1 should be moved to the materials and methods section, in a section called sample selection. In addition, it must be detailed if it was chosen at random or for convenience and how the calculation of the sample size was carried out so that the sample is representative of the population in which it is working.
4. Table 1, 2, 3 and 4. At the bottom of the table, I recommend that the statistical test used to determine statistical significance be expressed. Also, I suggest that numbers where statistical significance was found be in bold.
5. Multivariate regression analysis. I recommend that multivariate regression analysis be explained in more detail. It is necessary to explain what the dependent variable is, what the independent variables are, explain a bit the value of the OR and express what percentage of the variance this model affects.
6. Limitations of the study. I believe that an important limitation is the fact that the data has only been collected in a single hospital. Furthermore, if the sample was collected for convenience and was not representative of the study population, it should be expressed as a limitation.
7. Another important limitation that I can observe is the psychometric properties of the questionnaires used. If they have not used validated questionnaires for the Chinese population, this is also a limitation.
8. Strengths of the study. As important as the limitations are the strengths of the study, which it undoubtedly also has. Perhaps one of the strengths is that the satisfaction of the health personnel who cared for these women was taken into account. The regression study is also a strength as long as it is clearly explained, as it is perhaps the most important part of the investigation.
This was my evaluation.
Thank you
Kind regards
Author Response
Point 1: The introduction seemed adequate to me, but I recommend adding a couple of lines to the last paragraph explaining explicitly what the objective of this investigation was. This point was addressed in the abstract but, however, it was not expressed in the introduction and I consider it important.
Response 1: The aim of the study had been stated in the last paragraph of Introduction in the original manuscript. In order to further emphasize it, we now put it in a separate paragraph at the end of Introduction.
Point 2: It is necessary to explain in a more developed way which questionnaires were used to measure maternal satisfaction. There are already validated questionnaires in the literature that perfectly measure maternal satisfaction with childbirth. Some of them attend to the pain perceived by the women, to the attention received by the midwife, the obstetrician or the anesthesiologist. It would be necessary to know if some of these validated questionnaires have been used or self-made questionnaires have been used. In this case, it must be explained how the questionnaire was developed and validated to assess its psychometric properties.
Response 2: Although there were some validated questionnaires used to measure maternal satisfaction, few of them have been translated into Chinese and were suitable for this study. In this study, we conducted a satisfaction survey, rather than an official questionnaire. The satisfaction survey has been used in our hospital for over twenty years. The survey assesses the overall satisfaction on three aspects: the satisfaction score before epidural analgesia catheter insertion, the satisfaction score during labor , and the satisfaction score 24 hours postpartum. The satisfaction was scored on 5-point Likert scale (1: highly dissatisfied, 2: dissatisfied, 3: neutral, 4: satisfied, 5: highly satisfied). The survey questions were evaluated by our hospital experts to assess the suitability of the words and questions arrangement. However, an official CVI (Content Validity Index) score was not calculated. We explained this in “Materials and Methods” section (line 143 to line 152).
Point 3: The choice of participants must appear in the methodology section. The text written between lines 180 and 185 and figure 1 should be moved to the materials and methods section, in a section called sample selection. In addition, it must be detailed if it was chosen at random or for convenience and how the calculation of the sample size was carried out so that the sample is representative of the population in which it is working.
Response 3: A new section “Sample selection” is created in the paragraph of Materials and Methods (line 178 to 188), and Figure 1 is moved into it. The details of the sample size calculation is also documented inside. We included all the cases that got hospitalized and had epidural anesthesia for labor pain control during the study period as long as they did not fit into our exclusion criteria, therefore, the cases were not chosen at random and more like convinience sampling.
Point 4: Table 1, 2, 3 and 4. At the bottom of the table, I recommend that the statistical test used to determine statistical significance be expressed. Also, I suggest that numbers where statistical significance was found be in bold.
Response 4: We supplemented the statistical test informations in the tables, and emphasized the statistically significant numbers.
Point 5: Multivariate regression analysis. I recommend that multivariate regression analysis be explained in more detail. It is necessary to explain what the dependent variable is, what the independent variables are, explain a bit the value of the OR and express what percentage of the variance this model affects.
Response 5: We re-conducted regression analysis and adjusted our table 5. Details of the multivariate regression analysis are now provided in the “Statistical Methods” section.
Point 6: Limitations of the study. I believe that an important limitation is the fact that the data has only been collected in a single hospital. Furthermore, if the sample was collected for convenience and was not representative of the study population, it should be expressed as a limitation.
Response 6: We adjust the sentence stating our limitations in the last paragraph of Discussion to emphasize that the study was conducted in a single hospital and sampled by covinience sampling.
Point 7: Another important limitation that I can observe is the psychometric properties of the questionnaires used. If they have not used validated questionnaires for the Chinese population, this is also a limitation.
Response 7: We agree with the reviewer’s opinion and put this limitation in the last paragraph of Discussion in the revised version.
Point 8: Strengths of the study. As important as the limitations are the strengths of the study, which it undoubtedly also has. Perhaps one of the strengths is that the satisfaction of the health personnel who cared for these women was taken into account. The regression study is also a strength as long as it is clearly explained, as it is perhaps the most important part of the investigation.
Response 8: We explained the regression analysis section and hope to make it one of the strengths of this study.

Reviewer 2 Report
Thank you for the opportunity to review the manuscript Involving a Dedicated Epidural-Caring Nurse in Labor Ward Practice Improves Maternal Satisfaction towards Childbirth: a Retrospective Study , submitted to the Healthcare Journal. Congratulations for this interesting manuscript about this important issue.
The different sections of the study seem well addressed, although I offer some suggestions below.
1. In Abstract section, line 21 you write ".104 of them..." Please, don´t start the sentence with a number. Line 28, keywords must begin with a capital letter and must be separated by a semicolon.
2- In the methods section, Statistical Methods subheading, dependent variable must be well defined. Which was the dependent variable in the multivariate linear regression model?
3. In the results section, tables must be placed into the text, when data are been presented. In table 5 , the multivariate linear regression model should contain all variables used in the model.
4. The conclusion section should be stronger. Please make more emphasis on the results found in the study.
5. Most references are old. It is convenient to replace some of them with more recent references. Less than 50% of references are recent .
Author Response
Point 1: In Abstract section, line 21 you write ".104 of them..." Please, don´t start the sentence with a number. Line 28, keywords must begin with a capital letter and must be separated by a semicolon.
Response 1: We corrected the mistakes as recommanded.
Point 2: In the methods section, Statistical Methods subheading, dependent variable must be well defined. Which was the dependent variable in the multivariate linear regression model?
Response 2: We re-conducted regression analysis and provided detailed explanations of the variables used in the main text (2.6. Statistical Methods, line 193 to 199, and in Results section, line 253 to 262).
Point 3: In the results section, tables must be placed into the text, when data are been presented. In table 5 , the multivariate linear regression model should contain all variables used in the model.
Response 3: We re-conducted regression analysis and adjusted our table 5 and move the tables into the text.
Point 4: The conclusion section should be stronger. Please make more emphasis on the results found in the study.
Response 4: We re-wrote the conclusion to emphasize our research findings more stronly.
Point 5: Most references are old. It is convenient to replace some of them with more recent references. Less than 50% of references are recent .
Response 5: We replaced references 1, 12, 15 to more recent articles. We also added new referneces 16 and 17 into our article.

Round 2
Reviewer 1 Report
Dear authors,
It was my pleasure to review this improved version of the manuscript.
The authors have followed almost all the recommendations provided by the different reviewers, significantly improving the quality of the manuscript. Although some methodological aspects could have been improved from the beginning of the investigation, I think that in its current form it could be suitable for publication.
Kind regards